# Heavy-Ion Collisions toward High-Density Nuclear Matter

**DOI:** 10.3390/e24040482

**Published:** 2022-03-30

**Authors:** Shoji Nagamiya

**Affiliations:** 1KEK (High Energy Accelerator Research Organization), 1-1 Oho, Tsukuba-shi 305-0801, Japan; nagamiya@post.kek.jp; 2RIKEN (Institute for Physical and Chemical Research), 2-1 Hirosawa, Wako-shi 351-0198, Japan

**Keywords:** relativistic heavy-ion collision, high-temperature matter, high-density nuclear matter, RHIC, LHC, NICA, FAIR, HIAF, J-PARC-HI

## Abstract

In the present paper, the current efforts in heavy-ion collisions toward high-density nuclear matter will be discussed. First, the essential points learned from RHIC and LHC will be reviewed. Then, the present data from the STAR Beam Energy Scan are discussed. Finally, the current efforts, NICA, FAIR, HIAF, and J-PARC-HI (heavy ion) are described. In particular, the efforts of the J-PARC-HI project are described in detail.



**Prologue**



This paper is dedicated to the late Professor József Zimányi. I met with him, for the first time, in 1979 in Copenhagen when he was analyzing early-day pion data from our group at the Bevalac. We discussed a lot together. Four years later in 1983, we met again at Lake Balaton when the conference held there. At that time, Professor Zimányi invited me naturally to sing many songs together at the lakeside. Since then, both of us have sung together at many conferences. In 2005, when the Quark Matter Conference was held in Budapest, I met with him again together with his wife Magdolna (called Magda). A picture of two of them, which was taken at that Conference, is shown in Figure 1. Since then, Magda has contributed a lot in studying the history of Judo in Hungary together with me until she died in 2016, as my grandfather taught Judo in Hungary in 1905; his Judo book was discovered in Budapest in 2005. Soon after this photo was taken, Professor József Zimányi, unfortunately, passed away in 2006.

## 1. Introduction

Historically, the efforts on the studies of high-energy heavy-ion collisions (called relativistic heavy-ion collisions) started at the Bevalac (1974) at LBL up to 2 AGeV, where AGeV means GeV per nucleon, followed by the AGS (1986) up to 14 AGeV at BNL as well as by the SPS (1986) up to 200 AGeV at CERN. Later, SIS at GSI started in 1990 up to 2 AGeV. In 2000, the operation of RHIC up to 200 AGeV in a collider mode started at BNL, followed by LHC up to 5.5 ATeV in 2010, again, in a collider mode at CERN. By looking at these histories, extremely rapid progress has been achieved by thousands of physicists in relativistic heavy-ion collisions.

In the early universe, an accepted story is that light-mass quarks with masses of a few MeV were created immediately after the Big Bang. Then, at 10 μs after the Big Bang, the quarks were confined into nucleons (or, in general, hadrons) at temperatures of *T*~2 × 10^12^ K. At that time, the nucleon with a mass of ~1 GeV, which is significantly larger than the quark mass, was generated [1]. In relativistic heavy-ion collisions at RHIC or LHC, the colliding nuclei would penetrate through each other to create a hot and baryon-free region behind, as schematically illustrated in Figure 2 (right). In the middle region (or, in the mid-rapidity region), physicists believed that the system would be heated up well above *T*~2 × 10^12^ K and, therefore, an assembly of free quarks, antiquarks, and gluons could be created in a laboratory, as the case predicted in the early universe. The first experimental goal of the relativistic heavy-ion collisions at RHIC and LHC is to find a deconfined quark phase called the quark–gluon plasma. I will briefly review experimental progress in Section 2.

In the universe, hydrogen atoms were formed about 370 thousand years after the Big Bang. Then, these atoms were combined to form stars very slowly on the order of hundreds of million years or more. Inside the star, nuclear fusions would occur to form primarily ^4^He by emitting light together with heat from the star. Occasionally, in some big stars, nuclear fusions would proceed to form Ne or Mg atoms or even heavier Fe atoms. There, the inner core would be pressurized to induce the outer core to explode, which is known as a supernova explosion. Due to gravitational collapse, the neutron star was born by absorbing surrounding electrons by protons. The central density of the neutron star is of the order of 3–7 times the normal nuclear matter density. As described later in this study, such high-density matter would or could be created via relativistic heavy-ion collisions at 10–30 AGeV beams on a fixed target including the AGS energy region, as schematically illustrated in Figure 2 (left). Unfortunately, the AGS is not available now for experiments, so new efforts are currently in progress to attain these energies with new accelerators for the study of high-density matter. The effort toward high-density matter is the main topic of this paper, and it is described in Section 3 and Section 4.

Figure 3 is a well-cited phase diagram of the plane of temperature (*T*) and baryonic chemical potential (μ_B_), taken from the 2015 NSAC Long Range Plan, on which the coverage of various accelerators is superposed.

## 2. Results from RHIC and LHC

### 2.1. Initial Data from RHIC

Immediately after RHIC started its operation, two major discoveries were reported. The first was the high *p*ᴛ suppression [2] of particle production, as shown in Figure 4 (left), which strongly suggested that the particle energy would be lost when partons traversed through a deconfined quark matter named the quark–gluon plasma (QGP). Secondly, a strong elliptic flow v_2_ was observed, which agreed very much with a hydrodynamical flow with exceedingly small viscosity, as shown in Figure 4 (right) [3].

Namely, dense, and low viscous fluid seems to be created in nuclear collisions at RHIC. These two are called jet quenching and a strong elliptic flow, respectively.

If these two observations are associated with the formation of QGP, the initial temperature from the mid-rapidity region must exceed, at least, the critical temperature of 2 × 10^12^ K that was predicted by the lattice QCD results [4] (see also Section 1). Many people believe that the initial temperature achieved by RHIC can be observed through measurements of direct photons. Trials to measure virtual direct photons from e^+^e^−^ pairs were carried out [5] by showing the initial temperature was above 2 × 10^12^ K, although later studies of direct-photon flow effects have obscured this conclusion [6].

Therefore, the two unique observations in Figure 4 would likely be associated with the formation of QGP. In the following subsections, these two features are discussed in detail.

### 2.2. High p_T_ Suppression

At LHC, the *p*_T_ suppression has also been observed by covering a much broader range up to 100 GeV. To estimate an energy loss for a parton at RHIC and LHC, a theory group called the JET Collaboration extracted the jet transport parameter $\hat{q}$ [7],
$$\hat{q}\;=\;1.2\;+\;0.3\;{\mathrm{GeV}}^{2}/\mathrm{fm}\;\mathrm{in}\;\mathrm{Au}\;+\;\mathrm{Au}\;\mathrm{at}\;\sqrt{s_{NN}}\;=\;200\;\mathrm{GeV}\;(\mathrm{RHIC})$$
$$\hat{q}\;=\;1.9\;+\;0.7\;{\mathrm{GeV}}^{2}/\mathrm{fm}\;\mathrm{in}\;\mathrm{Pb}\;+\;\mathrm{Pb}\;\mathrm{at}\;\sqrt{s_{NN}}\;=\;2.76\;\mathrm{TeV}\;(\mathrm{LHC})$$
for a quark with initial energy of 10 GeV. Here, $\hat{q}$ is the rate of *p*_T_ broadening per unit length and is proportional to (*p*_T_)^2^. Since the radiative energy loss *dE*/*dx* is proportional to (*p*_T_)^2^, the energy loss for a parton traversing the length *L* is proportional to $\hat{q}$ × *L*.

When LHC started its operation, an interesting result, a large jet *E*_T_ asymmetry, was reported from the ATLAS group [8], as shown in Figure 5 (left). This asymmetry is direct evidence of the parton energy loss in the bulk QGP, similar to high-*p*_T_ suppression observed at both RHIC and LHC. In addition, if one looks at the lost jet energy distribution on the other side, which was also observed by the CMS [9] and STAR [10], the lost energy is distributed widely. These data support the high-*p*_T_ suppression, which is schematically illustrated in Figure 5 (right).

### 2.3. Quark Number Scaling and Perfect Fluidity of QGP

At RHIC, if all the available data of v_2_ flow values are normalized by the number of quarks *n*_q_, namely, two for mesons and three for baryons, then, the entire plots of v_2_/*n*_q_ vs. *KE*_T_/*n*_q_ seem to be on the same curve [11,12], as shown in Figure 6, where *KE*_T_ = (*m*^2^ + (*p*_T_)^2^)^1/2^ − *m.* This is called the quark number scaling and, again, this fact gives a strong hint on the formation of bulk QGP.

An interesting theoretical summary of the flow is shown in Figure 7 [13]. The values, of course, depend strongly on the initial condition. According to their detailed analysis and calculations, the viscosity divided by the entropy density, η/s, is 0.2 for LHC and 0.12 for RHIC, after fitting all the observed v_2_ to v_5_ values. Both values on η/s are extremely small and close to the non-viscous fluid limit of η/s = 1/4π = 0.08. Of course, these numbers, 0.2 and 0.12, are not exact and only indicative, but the result was amazing, because at both RHIC and LHC, the QGP, if formed, is almost a perfect liquid without any viscosities. This is quite different from the previously expected gas phase of QGP.

In Figure 8, the values of η/s are compared for several liquids, H_2_O, He, and QGP. The value for QGP is much smaller than in any other case. Additionally, among QGP, the η/s increases slightly as temperature increases from *T*_C_.

Intuitively, an extremely small viscosity is explained in the following way. Here, we are talking about shear viscosity. If partons are interacting strongly with each other, then a parton’s mean free path is short and, therefore, interactions are limited only among neighboring partons. Interactions of the original parton with partons far from it are, thus, exceedingly small. This fact causes an extremely small viscosity for the entire QGP system and induces extremely small shear viscosity. On the other hand, if partons are weakly interacting, then it means a parton carries a long mean free path. In this case, the information exchange occurs among partons at a long distance and, thus, induces a large viscosity. Therefore, small η/s means that partons are strongly interacting [14]. The coupling reaches the maximum at *T*_C_ and becomes slightly weaker at higher temperatures. This could be a reason why the value of η/s is slightly larger at LHC than at RHIC.

### 2.4. Debye Screening Effect (Melting of Quarkonium)

For a long time [15], a suppression of heavy quarkonium has been suggested as convincing evidence for the formation of QGP since the formation of mesons of quark and antiquark pairs, such as *c* or *b*, are suppressed due to the Debye screening effect of the QCD interactions. This Debye screening is also called the melting of quarkonium. Figure 9 shows the results, suggesting the melting of quarkonia. Data shown are from the CMS group at CERN-LHC [16,17].

### 2.5. Chemical Freezeout and Hadronization

Once the QGP was formed, another interesting subject is at what temperature quarks are combined into mesons and baryons. Recently, Andronic et al. [19] analyzed all the data from SIS, AGS, SPS, RHIC, and LHC and plotted the chemical freezeout temperatures, as shown in Figure 10, as the QGP would hadronize at the chemical freezeout temperature. From their studies, it was concluded that the chemical freezeout temperature at μ_B_ ≈ 0 is close to 156 MeV. Additionally, this number is close to the critical temperature calculated by the lattice QCD [20] of 155 ± 10 MeV, as indicated by the shaded area in Figure 10. Of course, many new calculations by the lattice QCD exist since the work in 2014 [20], for example, by the same Hot QCD Collaboration which obtained *T*c = 156.5 ± 1.5 MeV in 2019 [21]. 

### 2.6. Summary at RHIC and LHC

A summary of the knowledge on $\hat{q}$, η/s, and critical temperature, *T*_C_ from the lattice QCD results, is shown in Table 1, although other theoretical predictions may exist.

Recently, both RHIC and LHC communities presented their long range plans. Both operations are delayed but RHIC will be completed in 2025 by converting it to EIC, whereas the LHC heavy-ion program will continue to run after this date.

## 3. Efforts toward High-Density Nuclear Matter

### 3.1. Can High-Density Nuclear Matter Be Formed?

First, a comparison of baryon distributions among AGS, SPS, and RHIC energy regions is shown in Figure 11 [22]. In the central region, namely at around y_CM_ = 0, the density of baryons is almost zero at the RHIC energies at 200 AGeV in a collider mode, as intuitively explained in Figure 2. At the SPS energies of 200 AGeV on a fixed target, the density is finite in the mid-rapidity region, while the collision is still forward and backward peaked. Finally, at the AGS energies of 11–14 AGeV on a fixed target, the baryon density is peaked at y_CM_ = 0. The nucleons seem to stop each other in the center-of-mass frame at 10–20 AGeV.

We consider a model case first. If two rectangular objects move rapidly and meet each other to completely stop in the center-of-mass frame, as shown in Figure 12 (left), what would happen? Landau [23] predicted many years ago that two objects are shrunk by 1/γ_cm_, so that a high-density region would be created at the density of 2γ_cm_, where 2 comes from two objects and γ_cm_ is the Lorentz γ factor in the center-of-mass (c.m.) frame. Note that in the case of no collisions, the object simply rotates for observers without shrinking, as pointed out by Weisskopf [24].

More realistically, Goldhaber [25] thought of this problem in terms of nucleon–nucleon collisions, as shown in Figure 12 (right). In low energies at ~10–50 AMeV in the c.m. frame, each nucleon–nucleon collision is almost isotopic, so that nucleons would escape easily into free space. On the other hand, in high energies >1 AGeV in the c.m. frame, the colliding nucleons are forward peaked. Therefore, colliding nucleons are confined inside the nucleus. After a few collisions until nucleons almost stop each other, the nucleons are still confined inside the nucleus. Goldhaber’s conclusion was, therefore, the same as Landau’s prediction. At the AGS energy of 11–14 AGeV, γ_cm_ is about 3, so 6 times the normal density would be expected.

Before the J-PARC construction, a prototype project called the JHF was planned at KEK in Japan. Here, the accelerations of heavy-ions were planned. Therefore, Onishi [26] created a plot for the collision path at JHF at 25 AGeV heavy-ions, as shown in Figure 13. This hadron cascade calculation predicts that 9 times the normal nuclear matter density could be expected. The value of γ_cm_ in this case is ~3.5 so that 7ρ_0_ would be expected, while the calculated value is larger than this 7ρ_0_.

Concerning the high-density nuclear matter, intense theoretical studies are in progress on the neutron star. In the interior of the neutron star, in which high-density matter would exist, there are two possibilities. One is a normal strange hadronic matter, which is an assembly of p, n, Λ, and Ξ [27], whereas the other is a quark matter [28,29,30]. Regardless of whether hadronic matter or quark matter, a strangeness-rich object would be created and, therefore, the study of particles with strangeness is an interesting and important subject in the search of high-density matter. See also Section 3.4.

### 3.2. Search for Critical Point by the STAR Beam Energy Scan

The existing experiment toward high-density is the STAR Beam Energy Scan (STAR BES), which is lowering the RHIC beam energy as low as possible with a fixed target instead of a collider mode. The main purpose of this STAR BES is to cover the energy region between the AGS and SPS heavy-ion energy regions and even lower beam energies.

A phase diagram of nuclear matter in terms of temperature and baryonic chemical potential is shown, again, in Figure 14 [31]. In this figure, the baryonic chemical potential (μ_B_) is plotted in a log scale, while the temperature is on a linear scale, similarly to Figure 10. The coverage of the STAR BES is also presented in this figure. At RHIC and LHC, no sudden phase transitions were observed from the hadronic phase to QGP. This transition is conventionally called the crossover transition. On the other hand, the first-order phase transition might or would be expected at high density. When the first-order transition occurs, the starting point of the first-order transition is named the “critical point” [32,33]. (See the review of the QCD phase diagram on this point [33].) The first goal of the STAR BES was to explore if this critical point exists [31,34].

At the critical point, it is expected that a large fluctuation in the baryon number could occur. Namely, for the net proton number *N*, the ratio of the 4th order cumulant divided by the 2nd order cumulant, κσ^2^ must deviate sharply from 1. Here, κ is defined more precisely, κ = [<(δ*N*)^4^>/σ^4^]−3, where δ*N* = *N*−*M*, *M* is the mean and σ is the standard deviation. Figure 15 shows recent results from the STAR BES [34], which hint that the value of κσ^2^ sharply drops at √s_NN_ ≅ 20 GeV and increases rapidly toward √s_NN_ ≅ 8 GeV. If this fluctuation phenomena were due to the critical point, then it would be very interesting to study the region below √s_NN_ ≅ 8 GeV, namely, below ≅ 30 AGeV on a fixed target. The STAR group is now planning to cover lower energy regions, as shown by the FXT shown in both Figure 14 and Figure 15.

### 3.3. Bulk Properties (Chemical Equilibrium, Kinetic Equilibrium, Blast Wave Flow, and v_2_)

Other illuminating data from the STAR BES are shown in Figure 16. Figure 16 (left) shows the chemical freezeout temperature *T*_ch_, together with the kinetic freezeout temperature *T*_kin_ and a transverse blast wave flow velocity <β>. A method of how to decompose kinematical and blast waves has been known for a long time since the prediction by Siemens and Rasmussen [35] as well as from the data at the Bevalac [36]. Now, the value of *T*_ch_ stays almost constant at above √s_NN_ = 8 GeV. On the other hand, both *T*_ch_ and *T*_kin_ merge at around √s_NN_ = 6–8 GeV, and they sharply drop below this energy region. In addition, the value of <β> sharply drops below √s_NN_ = 6–8 GeV. The other data, shown in Figure 16 (right) [36], also show a sudden drop in the v_2_ value below √s_NN_ = 6–8 GeV. The elliptical flow even changes its sign from plus to minus at around √s_NN_ = 3–4 GeV.

If the critical point exists at around √s_NN_ = 8–20 GeV, as hinted by the fluctuation data shown in Figure 15, the first-order transition could occur below √s_NN_~8 GeV, so it is interesting to study this energy region more carefully in the future, although the physical interpretation of Figure 16 is yet unclear.

### 3.4. Strangeness Production and Formation of Hypernuclei

The third point is strangeness enhancement at around √s_NN_ = 8 GeV, as shown in Figure 17 (left). There, K^+^/π^+^ ratios are enhanced as compared to K^−^/π^−^ ratios [37]. At the AGS, initial studies on this point were performed for the first time [39]. These phenomena were interpreted by a thermal model [40], but a peak at around √s_NN_ = 8 GeV suggests a change of freedom, either baryonic to mesonic freezeout [41] or hadrons to QGP [42]. This would be an important aspect when one plans experiments in the compressed baryon matter [31].

On the other hand, the production of Λ is also enhanced at √s_NN_~8 GeV. This implies that the formation of hypernuclei would be enhanced as well at this energy region. Andronic et al. [43] calculated production rates in heavy-ion collisions, based on a coalescence model. The results are shown in Figure 17 (right). Note that the production of hypernuclei with multi-strange quarks at strangeness equal or larger than 3, such as _ΛAΞ_He (strangeness of −4), is possible by using heavy-ion beams alone, although high statistics are needed. As a byproduct of heavy-ion collisions in these energy regions, it would be interesting to study hadrons, hadron pairs, and hypernuclei with strangeness equal to or larger than 3.

### 3.5. Crossover Transition vs. First-Order Phase Transition

At this point, let us explain the difference between the crossover transition and the first-order transition. The entropy per volume increases rapidly in the first-order transition, while it increases gradually and smoothly in the crossover transition, as illustrated in Figure 18 (right). At a high-temperature region at zero baryon density, it has been confirmed that the crossover transition occurs, while at a high-density region no one knows if the transition is via the first order or by the crossover. Fukushima and Hatsuda [33] proposed a phase diagram, as shown in Figure 19. In this diagram, “Gas-Liquid” is written, which is a well-known liquid–gas phase transition [44]. The normal nucleus is regarded as a liquid, while if it were heated up to *T*~10 MeV, the nucleus would expand and dissolve into individual nucleons, and these nucleons behave like a gas. This transition is illustrated there. On the other hand, at the high-density region, Hatsuda et al. [45] recently proposed that the transition from the normal matter density to higher density must be the crossover transition, at least in the small temperature region, as illustrated in Figure 18 (bottom) as well as in Figure 19. This point must be studied experimentally in the future.

## 4. New Accelerator Projects toward High-Density Matter

### 4.1. Current and Future Accelerator Projects in the World

A summary of the current and future accelerators in the low-energy region toward high-density regions is illustrated in Figure 20 [47]. The STAR FXT, as well as the NICA collider, cover relatively higher energy regions while at low luminosities. On the other hand, FAIR, HIAF, and J-PARC-HI cover higher interaction regions due to the high-flux accelerators on a fixed target. The reaction rate reaches up to 10 MHz.

### 4.2. NICA

In Russia, the NICA project will be completed in 2023 by using 4.5 AGeV Nucleotron beams as an injector. A sketch from the NICA homepage is shown in Figure 21 [48]. Energy up to √s_NN_ = 11 GeV can be attained. The experiment called the Multi-Purpose Detector is scheduled to start operations by the time the collider starts its operation. The results will attract physicists in the field.

This successful project reminded me of the past proposal at KEK to inject from the 12 GeV proton synchrotron (3–6 AGeV heavy-ions) to the PS-Collider [49], which was proposed over 30 years ago to study high-density matter. Unfortunately, this project was not approved. Therefore, people at that time were also anxious about the NICA results.

### 4.3. FAIR

The most realistic approach in these energy regions is the FAIR Project at GSI. Figure 22 shows the current accelerator plan of SIS100 [50]. The proton equivalent energy is 29 GeV so 5–14 AGeV heavy-ion beams are available. The beam energy depends on the charge states of ions.

Two major experiments [51] are planned at SIS100, as shown in Figure 23. The first, called the CBM experiment, is to seek high-density matter from the participant regions of heavy-ion collisions. In the plane of *T* (temperature) and ρ (density), the collision will sweep both high density and high (but lower than at RHIC and LHC) temperature regions, as illustrated in Figure 23. It is possible that the system might dive into a new phase through the first-order phase transition. The other experiment, called the NUSTAR experiment, is to utilize the projectile spectators as beams. If incoming original beams are uranium, the neutron to proton ratio is about 1.6, whereas the neutron to proton ratio is almost 1 for light nuclei. Therefore, these fragments will create neutron-rich isotope beams. The latter type of study is already in progress in many facilities in the world.

### 4.4. HIAF

In China, the HIAF project is in progress, as shown in Figure 24 [52]. The project has many different goals. The main ring can accelerate proton beams up to 10 GeV, and thus, 1.7 AGeV for Kr beams. This is similar to the SIS or Bevalac beforehand, but it has the high-intensity capability to allow detailed studies on nuclear matter. Several versions of upgrade plans also exist there, as the space is unlimited.

This HIAF facility has another important goal. Near to this facility a big ADS proton driver called CiADS (China Initiative Accelerator Driven System) exists [53], at which extremely high-flux proton beams up to 10 mA can be accelerated. By using target fragments in pA collisions, as shown in Figure 24, they can use high-flux neutron-rich isotope ions as ion sources for the HIAF, as in the case seen in the ISOL facility at CERN.

### 4.5. J-PARC-HI

The J-PARC is a proton accelerator that provides high-flux beams at 750 kW for 3 GeV at RCS ring and 500 kW for 30 GeV at the main ring (MR). This J-PARC was constructed by a joint effort between two organizations, KEK and JAEA (Japan Atomic Energy Agency), and it has been delivering beams since 2009. The idea to accelerate heavy-ion beams at J-PARC was born about 10 years ago. This is called the J-PARC Heavy Ion Facility, abbreviated the J-PARC-HI [54]. The accelerator group has investigated accelerating heavy-ion beams up to uranium at high intensities, by constructing linac and booster rings, as shown in Figure 25.

Expected beam energy for MR for U beams is about 11 GeV, though it may be increased higher with an additional effort. Required for the J-PARC-HI is a heavy-ion injector alone, which is a tiny portion of the entire facility of the J-PARC. The beam intensity for heavy-ions is limited by either an injector intensity or the counting rates of detectors in the experimental device. Currently, with this configuration, an expected beam intensity is 10^11^ beams/s.

The current concern, however, is on the construction budget. It would cost around USD 150M for a heavy-ion injector accelerator alone. After looking at the budget in detail, including real estate and buildings, a significantly different idea has been newly considered since the beginning of 2020 in order to save money. This revised proposal is the following. The original linac must be replaced by a proposed 10 AMeV linac that was already requested by the JAEA to the Government, as a replacement for the existing Tandem accelerator at JAEA. In addition, the booster will be replaced by an already available booster ring, which was used in the past as a part of disposed 12 GeV KEK-PS system. Fortunately, this booster is still working for another purpose, even though the entire system was officially disposed. By applying these replacements to the original idea, the hardware cost would be reduced to less than USD 30M (20M for linac and 10M for high-vacuum booster). The only shortcoming is that the goal intensity is significantly reduced to the level of 10^8^ heavy-ion beams/s.

Figure 26 (**upper**) is the currently proposed linac by JAEA. Additionally, shown in Figure 26 (**lower**) is the KEK-PS Booster ring, which will be abandoned soon if not being used for any other purposes. The energy of heavy-ion beams at the exit of the booster is well over 100 AMeV (for protons 500 MeV), which is sufficiently high as an injector to the existing RCS ring at J-PARC. Of course, this booster can accept linac beams at 10AMeV. Although the beam intensity drops significantly, it is much better to start the heavy-ion project than to wait.

An experimental proposal [55] has already been submitted to the J-PARC PAC. In this experiment, the usage of the existing detector, the E16 Experiment, with a small modification was proposed. Namely, the existing beamline together with the existing experimental device was planned as a first step. Figure 27 shows the working E16 experiment at J-PARC, where electron pairs are being measured from primary protons to study if distortion occurs for the φ meson region as well as in other regions. In the future, other new experiments will be considered in heavy-ion experiments, for example, baryon-baryon interactions, hypernuclei, fluctuations, QGP search at high baryon density, etc.

## 5. Final Remarks

The summary of the paper is as follows. RHIC and LHC have played an extremely important role in exploring the existence of QGP as well as studying the properties of QGP. RHIC will complete its scientific mission by 2025. LHC has added and will add many more surprises by future unexpected discoveries for at least ~10 years.

The STAR BES opened new windows toward high-density regions. The extension of this project (FET) is in progress. Concerning the search for high-density nuclear matter, FAIR and NICA are progressing well. HIAF is also moving forward. J-PARC HI is now trying to meet the challenge of creating a new revised version for heavy-ion acceleration at J-PARC.

## Figures and Tables

**Figure 1 entropy-24-00482-f001:**
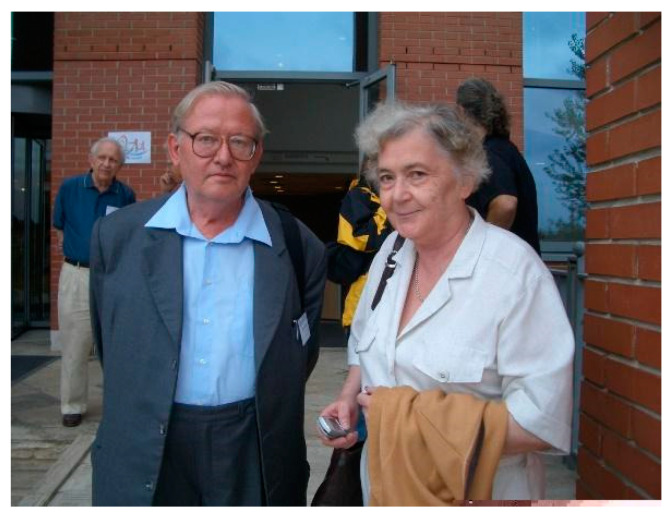
A photo taken in 2005 for Professor ZIMÁNYI József (1931–2006) and Ms. ZIMÁNYI Magdolna.

**Figure 2 entropy-24-00482-f002:**
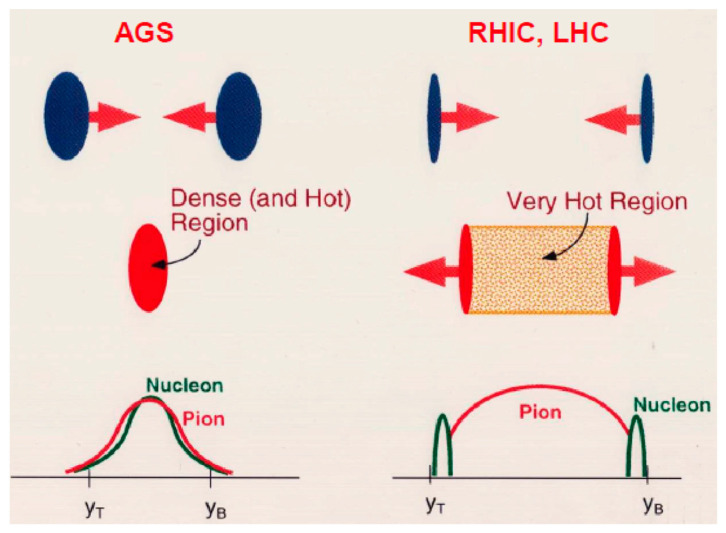
Comparison of relativistic heavy-ion collisions at the AGS energy region (**left**) and the RHIC/LHC energy region (**right**). At the RHIC/LHC energy region, the Lorenz-contracted nuclei penetrate through each other to form a hot but baryon-free region in the middle, whereas at the AGS energy region, colliding nuclei stop each other to form a baryon-rich region in the middle. y_T_ and y_B_ indicate target rapidity and projectile rapidity, respectively.

**Figure 3 entropy-24-00482-f003:**
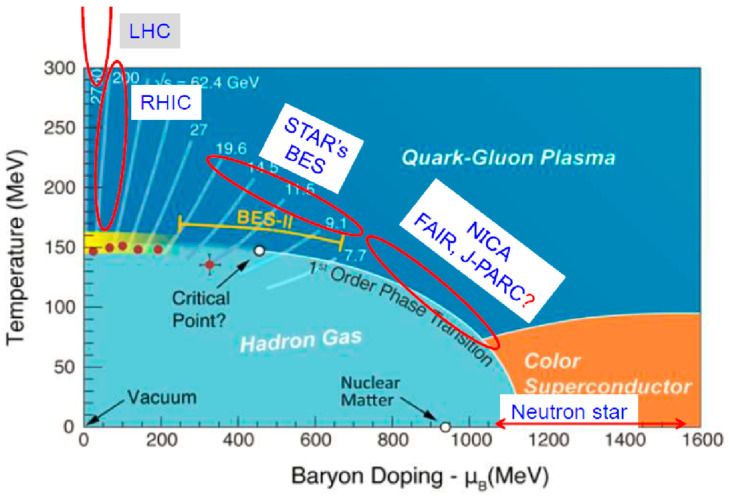
The proposed phase diagram of nuclear matter in terms of temperature (*T*) and baryonic chemical potential (μ_B_) where μ_B_ is a measure of density at above μ_B_ ~1 GeV.

**Figure 4 entropy-24-00482-f004:**
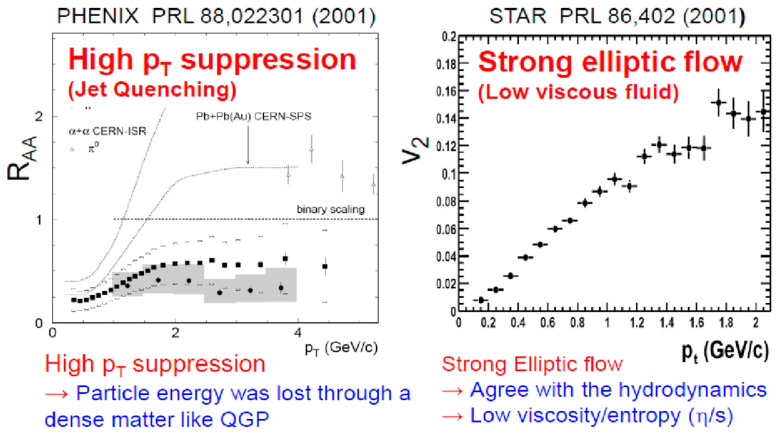
Jet quenching (**left**) [2] and strong elliptic flow (**right**) [3] observed at RHIC.

**Figure 5 entropy-24-00482-f005:**
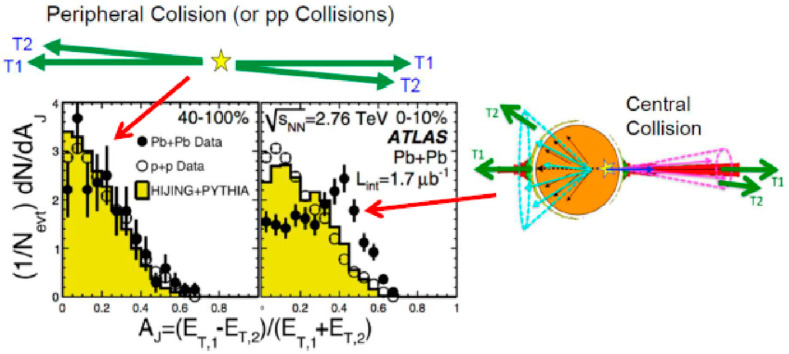
Azimuthal asymmetry of transverse energies observed in central collisions at ATLAS [8]. A similar phenomenon is observed at both CMS at CERN [9] as well as STAR [10]. Pictorial explanations for both peripheral and central collisions are also shown in the figure.

**Figure 6 entropy-24-00482-f006:**
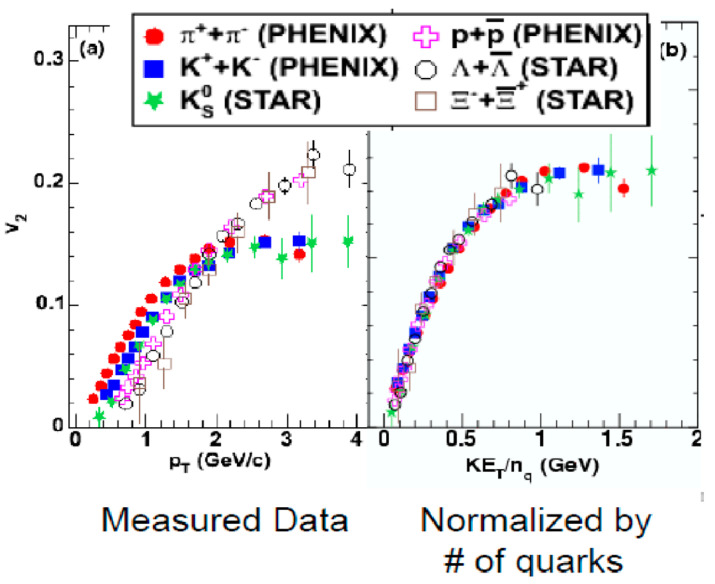
Quark number scaling of the v_2_ flow data: (**a**) measured and (**b**) normalized by *n*_q_ [11,12].

**Figure 7 entropy-24-00482-f007:**
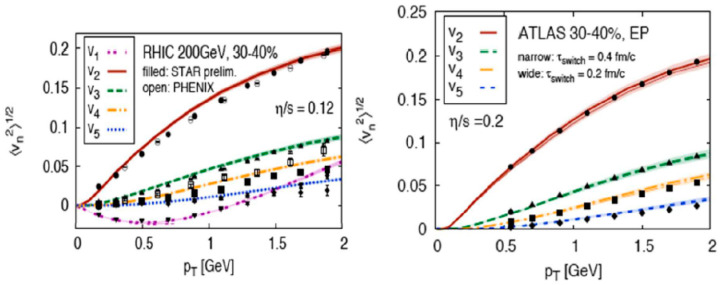
Analysis for the available data of v_2_ to v_5_ [13].

**Figure 8 entropy-24-00482-f008:**
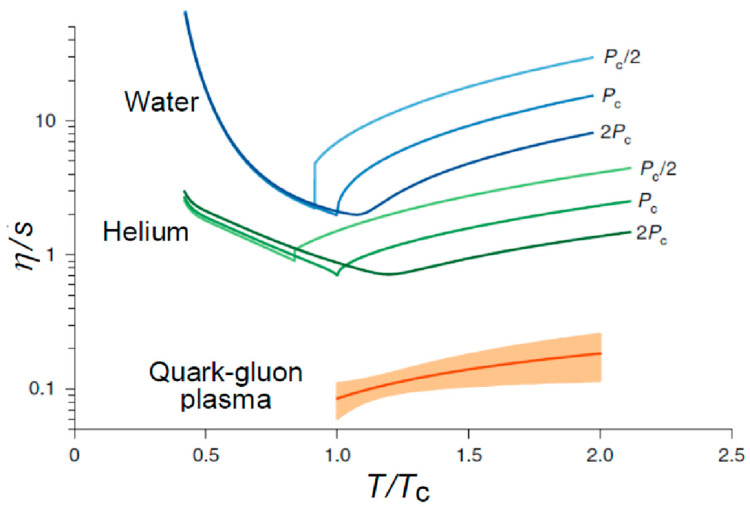
Comparison of η/s for various liquids. The figure is taken from [14].

**Figure 9 entropy-24-00482-f009:**
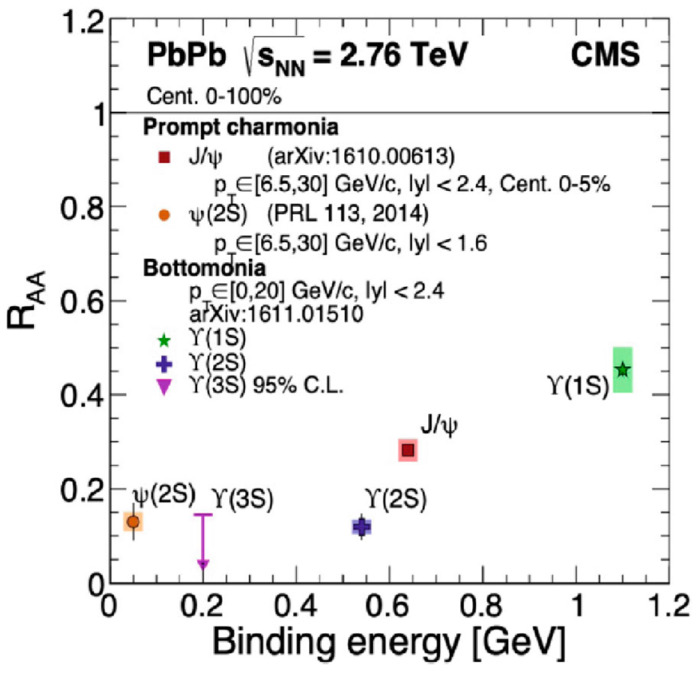
Suppressions of *J/*ψ (*c*$\bar{c}$) and Υ(upsilon = *b*$\overset{⥪}{b}$) and their excited states observed by the CMS group at CERN-LHC [16,17]. The figure is taken from [18].

**Figure 10 entropy-24-00482-f010:**
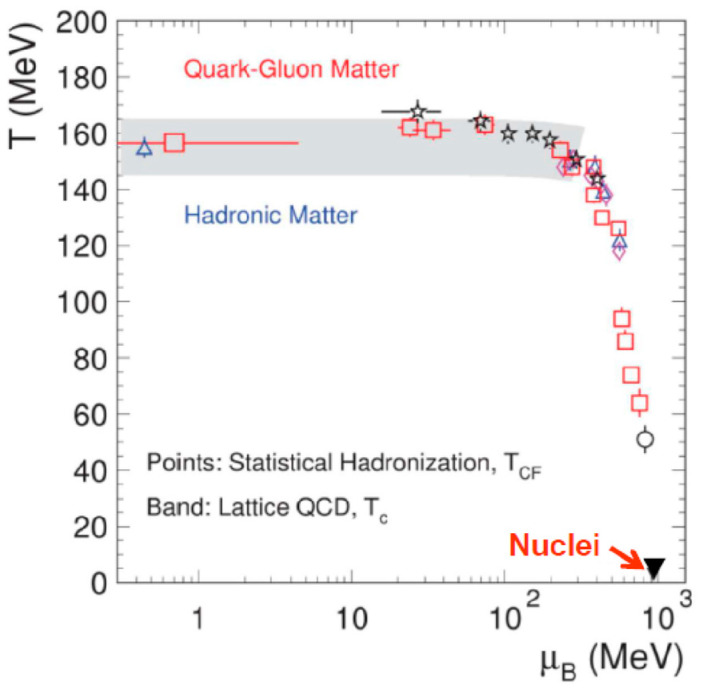
Chemical freezeout temperature obtained from data from SIS, AGS, SPS, RHIC, and LHC [19], as compared with the Lattice QCD result [20] indicated by the shadow area.

**Figure 11 entropy-24-00482-f011:**
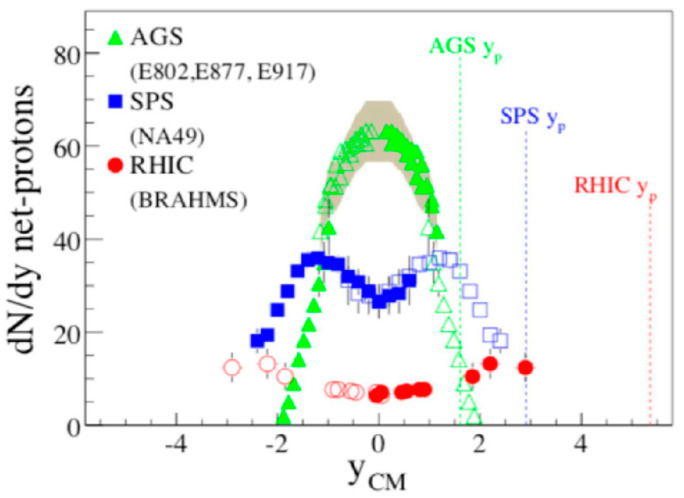
Net proton (proton–antiproton) distributions at AGS, SPS, and RHIC [22].

**Figure 12 entropy-24-00482-f012:**
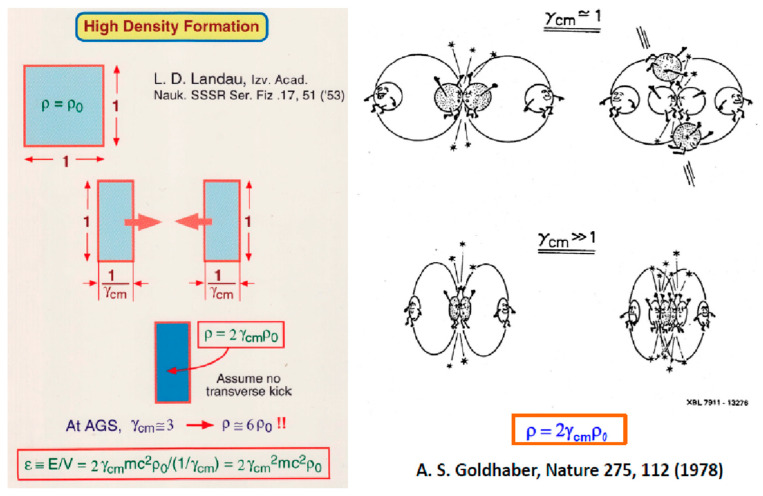
Schematic drawing when two colliding objects stop one another completely. **Left** figure is from Landau [23] and the **right** figure is from Goldhaber [25], though the illustrations are pictorial.

**Figure 13 entropy-24-00482-f013:**
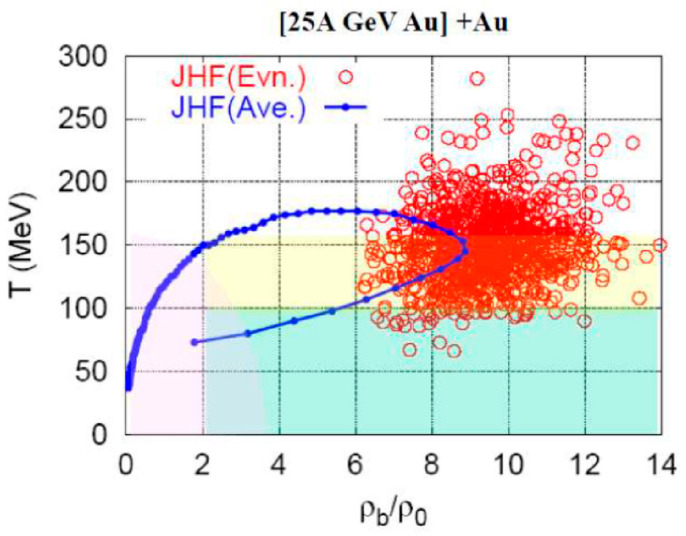
A hadron cascade calculation for heavy-ion collisions at 25 AGeV was created at the proposed JHF project in Japan (before J-PARC). At those energies, the collision reaches well above the expected value of 7ρ_0_ [26].

**Figure 14 entropy-24-00482-f014:**
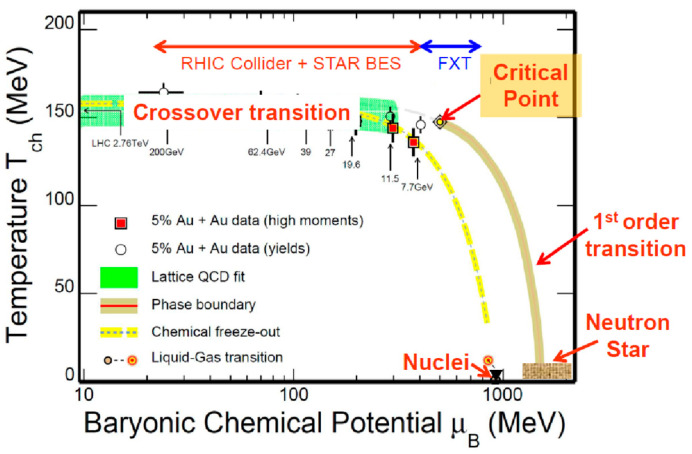
RHIC Collider, STAR Beam Energy Scan (BES), and its extension named FXT superposed on the phase diagram. It is a common expectation that the first-order phase transition would start from the critical point toward high-density regions [31,34].

**Figure 15 entropy-24-00482-f015:**
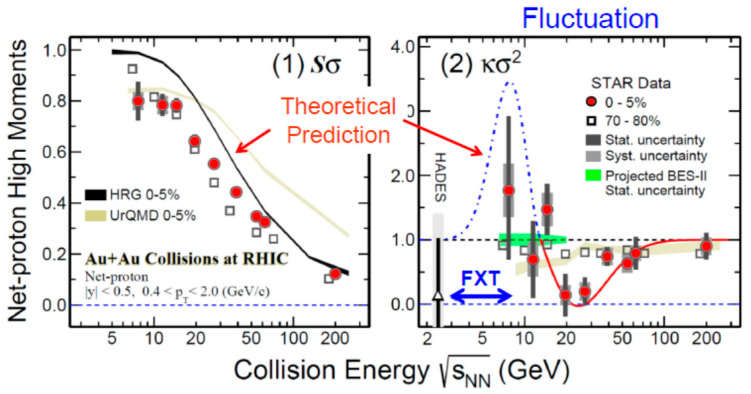
New data on fluctuations κσ^2^ (**right**) at the critical point between the crossover transition and the first-order phase transition [34]. Statistics are still low, though. The figure on the **left** is a normalized baryon number.

**Figure 16 entropy-24-00482-f016:**
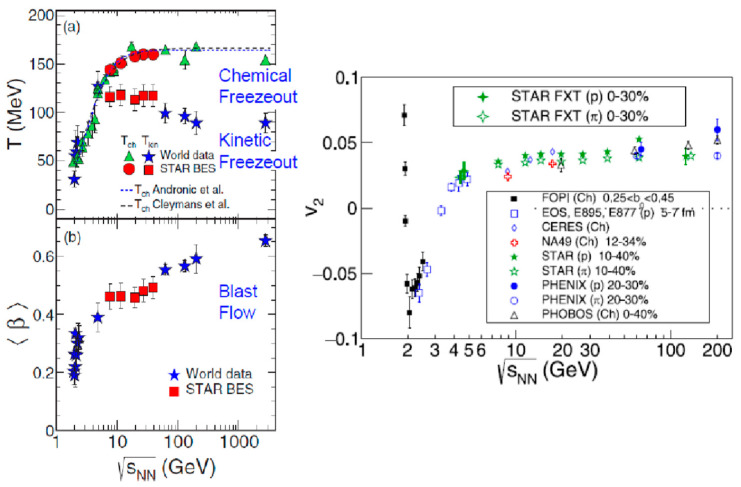
In the **left** figure, (**a**) shows chemical freezeout temperature and kinetic freezeout temperature. (**b**) indicates an extracted blast flow [37]. The **right** figure shows the observed change in v_2_ [38]. Note that at around √s_NN_ = 6–8 GeV, a sudden decrease occurs for *T*_ch_, *T*_kin_, <β>, and v_2_.

**Figure 17 entropy-24-00482-f017:**
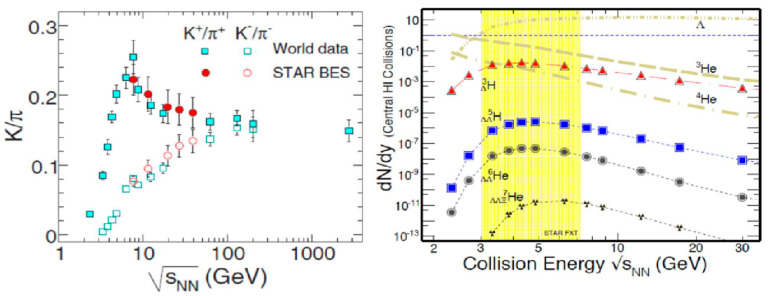
The observed K/π ratios (**left**) [37] and expected formation of hypernuclei in relativistic heavy-ion collisions (**right**) [43]. The latter is based on the coalescence model.

**Figure 18 entropy-24-00482-f018:**
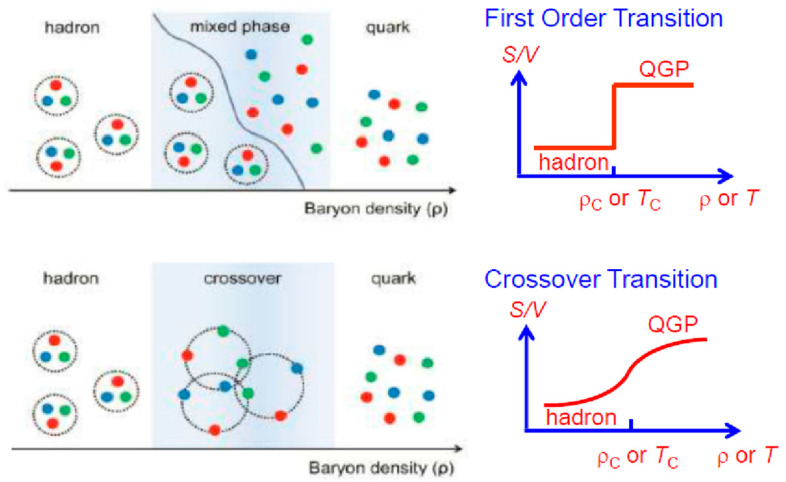
Schematic explanation of the first-order phase transition and the crossover transition. Figures on the **left**-hand side were taken from [46]. The **right**-hand figure is a pictorial explanation.

**Figure 19 entropy-24-00482-f019:**
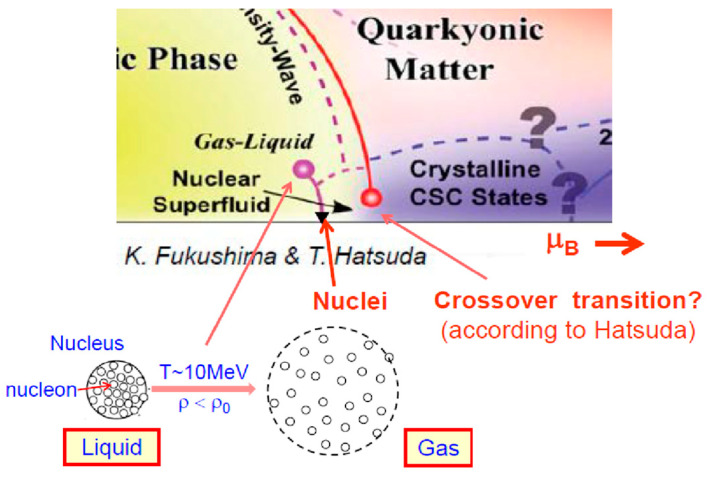
A proposed phase diagram of high-density nuclear matter [33]. The crossover transition was proposed by Hatsuda et al. [45], and it is illustrated in the previous Figure 18.

**Figure 20 entropy-24-00482-f020:**
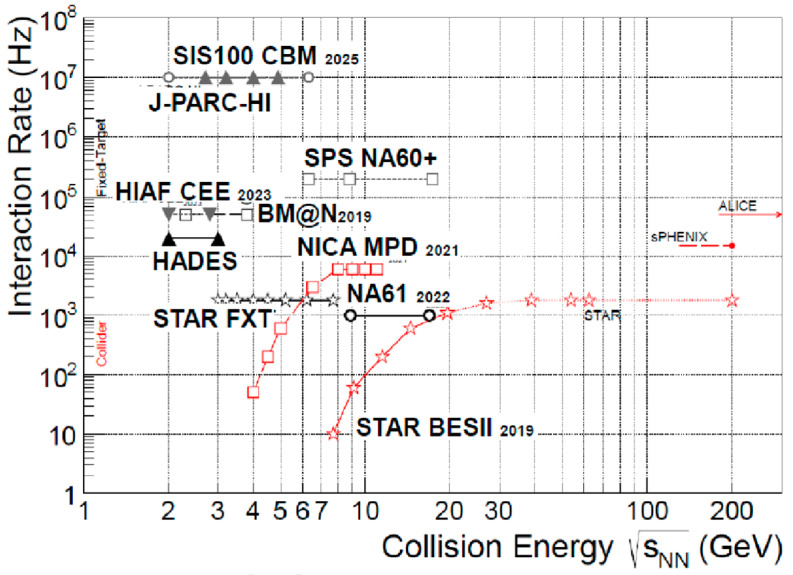
Planned future accelerators toward high-density nuclear matter. A high flux up to 10 Mz interaction rates is planned [47]. Only STAR BES has been running. Other are the future plans.

**Figure 21 entropy-24-00482-f021:**
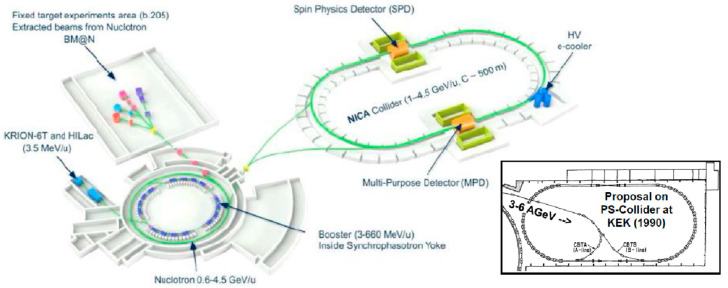
NICA project, which was completed in 2020. Beams from the Nucleotron at 4.5 AGeV are injected into a collider called NICA. Figure is taken from [48]. In the inset, a proposed project from 1990, called the PS-Collider [49] at KEK, is shown, although this project was not approved.

**Figure 22 entropy-24-00482-f022:**
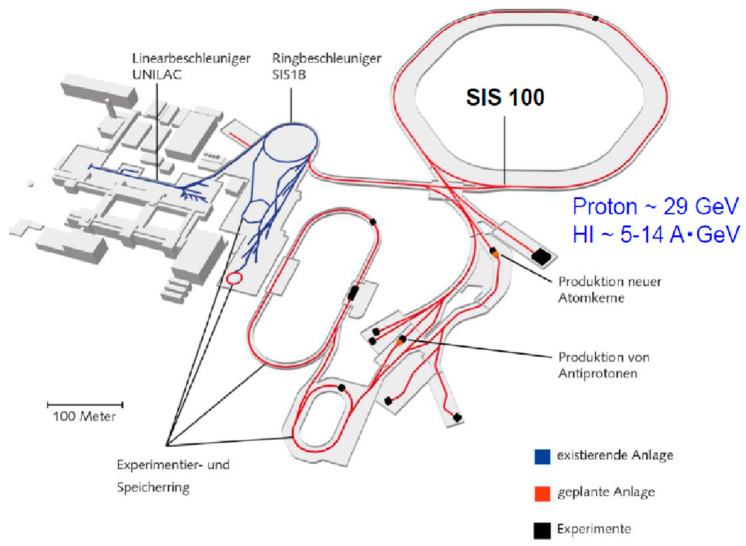
The FAIR accelerator project at GSI. The SIS 18 will be used as an injector. The facility will be completed by 2025. The figure is taken from [50].

**Figure 23 entropy-24-00482-f023:**
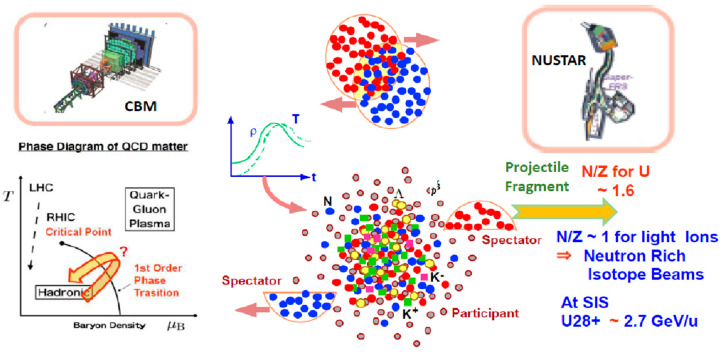
Two major experiments, CBM and NUSTAR, are planned at FAIR at CSI [51].

**Figure 24 entropy-24-00482-f024:**
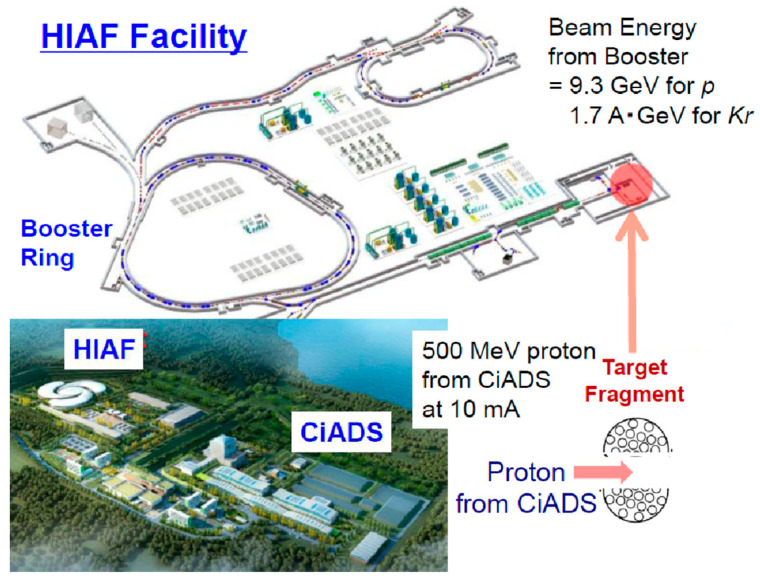
The HIAF facility for the study of nuclear matter (above) [51]. In parallel to this HIAF, extremely high-flux CiADS is being constructed. HIAF also uses neutron-rich isotopes by utilizing high-flux beams from CiADS as an ion source, as with ISOLDE at CERN, to significantly enrich the HIAF project [52].

**Figure 25 entropy-24-00482-f025:**
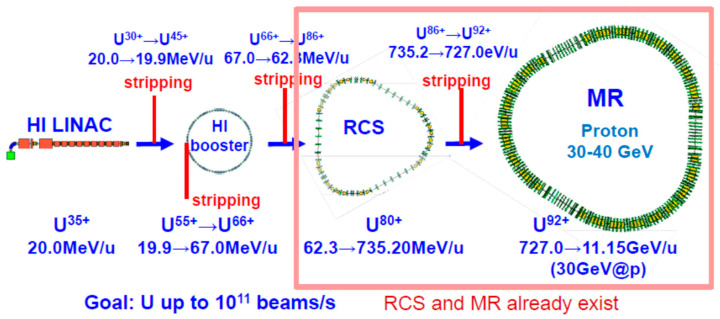
A J-PARC heavy-ion project in Japan, that requires heavy-ion injector alone, as the 3 GeV RCS and the 30–40 GeV MR already exist [54].

**Figure 26 entropy-24-00482-f026:**
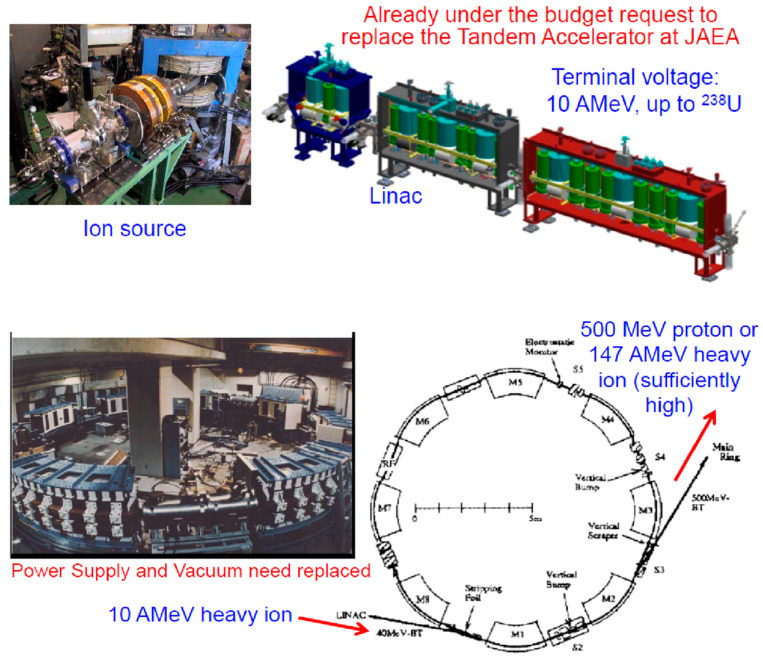
The **upper** figure is the proposed linac from JAEA and the **lower** figure is a planned usage of the existing KEK-PS Booster ring (already disposed but working at KEK). This revised plan was proposed in 2020.

**Figure 27 entropy-24-00482-f027:**
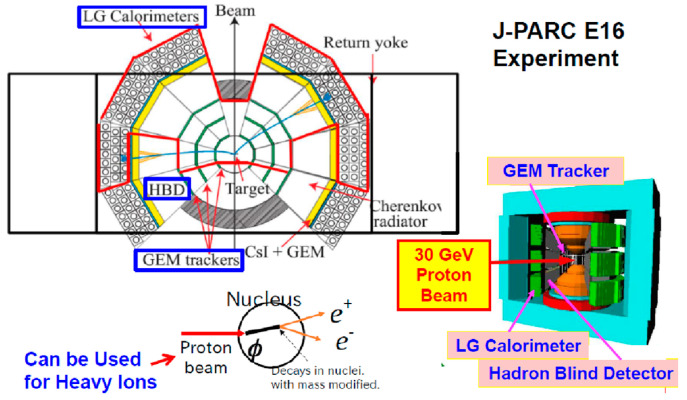
An initial plan for the J-PARC by using the existing E16 experiment [55].

**Table 1 entropy-24-00482-t001:** Summary of the knowledge on $\hat{q}$, η/s, and critical temperature, *T*_C_.

	$\hat{q}$ (GeV^2^/fm)	η/s	*T* _C_
RHIC 200 AGeV	1.2 ± 0.3	0.12	~156 MeV
LHC 2.67 ATeV	1.9 ± 0.7	0.2	

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
