# Peer review of "Heavy-Ion Collisions toward High-Density Nuclear Matter"

_entropy, 2022, doi:10.3390/e24040482_

Round 1

Reviewer 1 Report

It's a very well written review and I hope it will be very useful, specially for the newcomers in the field of heavy ion. It will serve as a nice reference, particularly to those who are looking towards future high density nuclear matter experiments and for a bit of historical background.

Author Response

I thank the referee for the report, and am grateful that the referee recommends that the manuscript be accepted for publication in Entropy.

Reviewer 2 Report

Nicely witten paper. Interesting review of the current efforts in heavy ion collisions toward QGP, worth publishing. I highly recommended manuscript for publication.

Author Response

(The authors gave the same response as above.)

Reviewer 3 Report

The paper under review represents a very nice review of the history of heavy-ion experiments and the efforts to map the phase diagram of QCD. I warmly recommend it for publications once the issues listed below have been properly resolved.

-p.2 last paragraph: the description of neutron star formation, "electrons would be combined with protons in the core nuclei by forming neutron 67 stars. " is perhaps a bit misleading, as neutron stars only form in the gravitational collapse after a supernova explosion

-p.3, first paragraphs of 2.1: I don't quite understand the sentence "particle energy would be lost when partons traverse through a dense deconfined quark matter if the quark-gluon plasma (QGP) were formed. ". How does the author define quark matter and quark-gluon plasma, and what separates the two concepts here?

-p.4, beginning of 2.2: if the figures "370 MeV" and "470 MeV" refer to temperatures, this should be clearly indicated. Note also a consistent typographical problem with the q hat parameter.

-p.5, first paragraph: the viscosity to entropy values of 0.2 and 0.12 are quoted with no uncertainties and no discussion of the subtleties related to their determination. It might be good to indicate to the reader that these numbers are not exact but only indicative.

-p.7, first line: the uncertainty of the critical temperature seems quite large to me, and only one reference is given which is close to 10 years old. This discussion could be extended a bit and at least one more recent paper (perhaps from the Wuppertal-Budapest group) cited. Also, note that T_c value in Table 1 is somewhat different from that discussed in the text

-p.10, first line: the author states that the presence of a critical point and a line of first-order transitions starting from it is expected. This is a very nontrivial claim, and would need further discussion. Historically, a first order transition was also expected based on mean field calculations at high temperatures, but lattice QCD has later shown the transition to be a crossover. Why could the same not happen at high density?

-p.12, first paragraph: I find this discussion very confusing. Regarding the text "Namely, starting from the normal nucleus the first phase transition would occur from liquid to gas transition, since the normal nucleus behaves like a liquid while at above 10 MeV the system behaves like a gas of nucleons. The phenomenon is called the liquid-gas phase transition [43]. After this point is passed, a gas of nucleons would experience a transition toward higher density from nucleon gas to the neutron star. According to Hatsuda, et al. [44], the transition from the hadron gas to neutron star could be the crossover transition at low temperatures. as illustrated in Figure 18 (left) and Figure 19. Namely, a quark matter is gradually formed. These points must be studied experimentally in the future. " I have the following questions/comments:
-What does "normal nucleus" mean here? This might be just a language issue, but this concept is not clear at all.
-What does "at above 10 MeV" refer to? Temperature perhaps, or some chemical potential difference?
-In the liquid gas phase transition one generally moves from a gaseous phase of hadrons at low density to a liquid phase of nuclear matter at higher density. This seems to be at odds with that is written here.
-What does "nucleon gas to neutron star" mean? This sounds like a description of the liquid-gas transition if neutron star refers to nuclear matter, but the wording "neutron star" is very nonstandard. Inside a neutron star, several different phases of matter are present simultaneously corresponding to different layers of the star.

Author Response

Thank you for your comments, please see the attachment

Round 2

Reviewer 3 Report

The author has improved the quality of the paper considerably, and I am happy to recommend it for publication.